# Estimation of Climatologies of Average Monthly Air Temperature over Mongolia Using MODIS Land Surface Temperature (LST) Time Series and Machine Learning Techniques

**Munkhdulam Otgonbayar [1], Clement Atzberger [2],\*, Matteo Mattiuzzi [3]
and Avirmed Erdenedalai [1]**

[1]   Institute of Geography and Geoecology, Mongolian Academy of Sciences, Ulaanbaatar 15170, Mongolia;
     munkhdulamo@mas.ac.mn (M.O.); abirmede@mas.ac.mn (A.E.)
[2]   Institute of Surveying, Remote Sensing and Land Information (IVFL), University of Natural Resources and
     Life Sciences (BOKU), 1190 Vienna, Austria
[3]   European Environmental Agency, 1050 Copenhagen, Denmark
\*   Correspondence:  clement.atzberger@boku.ac.at; Tel.: +43-1-47654-85700

**Abstract:** The objective of this research was to develop a robust statistical model to estimate climatologies (2002–2017) of monthly average near-surface air temperature (Ta) over Mongolia using Moderate Resolution Imaging Spectroradiometer (MODIS) land surface temperature (LST) time series products and terrain parameters. Two regression models were analyzed in this study linking automatic weather station data (Ta) with Earth observation (EO) images: partial least squares (PLS) and random forest (RF). Both models were trained to predict Ta climatologies for each of the twelve months, using up to 17 variables as predictors. The models were applied to the entire land surface of Mongolia, the eighteenth largest but most sparsely populated country in the world. Twelve of the predictor variables were derived from the LST time series products of the Terra MODIS satellite. The LST MOD11A2 (collection 6) products provided thermal information at a spatial resolution of 1 km and with 8-day temporal resolution from 2002 to 2017. Three terrain variables, namely, elevation, slope, and aspect, were extracted using a Shuttle Radar Topography Mission (SRTM) digital elevation model (DEM), and two variables describing the geographical location of weather stations were extracted from vector data. For training, a total of 8544 meteorological data points from 63 automatic weather stations were used covering the same period as MODIS LST products. The PLS regression resulted in a coefficient of determination ($R^2$) between 0.74 and 0.87 and a root-mean-square error (RMSE) from 1.20 °C to 2.19 °C between measured and estimated monthly Ta. The non-linear RF regression yielded even more accurate results with $R^2$ in the range from 0.82 to 0.95 and RMSE from 0.84 °C to 1.93 °C. Using RF, the two best modeled months were July and August and the two worst months were January and February. The four most predictive variables were day/nighttime LST, elevation, and latitude. Using the developed RF models, spatial maps of the monthly average Ta at a spatial resolution of 1 km were generated for Mongolia (~1566 × $10^6$ km²). This spatial dataset might be useful for various environmental applications. The method is transparent and relatively easy to implement.

**Keywords:** near-surface air temperature; MODIS land surface temperature; terrain parameters; partial least square regression; random forest regression

## 1. Introduction

Near-surface air temperature (Ta) is a key descriptor of the climate [1]. Ta is a critical variable to the effective understanding of the many physical and biological processes between the atmosphere and land systems [2–4] because it regulates many land surface processes such as photosynthesis, respiration, and evaporation [5]. As air temperature influences nearly all biotic processes [6], climatologies of Ta also permit a good characterization of terrestrial environmental conditions [5,7]. As this variable can change quickly over space, cost-efficient mapping procedures are needed that can depict Ta using high spatial resolution.

Since the early 1980s, various interpolation methods have been used to estimate Ta given adequate sample points [8,9]. The literature shows that the most common interpolation techniques are global interpolators, thin plate smoothing splines and different forms of kriging [10,11], inverse distance weighting [12], and climatologically-aided interpolation [13]. In the comparative study of [14], most interpolation methods gave similar results. However, interpolation errors typically range between 1 and 3 °C [15,16] depending on the spatial and temporal resolution of recorded Ta data and the density of the station network [17].

Direct measurements of Ta at a height of 2 m above ground are only available from a limited number of meteorological stations. In many cases such as Mongolia, the spatial coverage of these measurements is inadequate; in addition, typical Ta time series come with many missing values [6]. On the contrary, satellite-derived land surface temperature (LST) data are continuous in both spatial–temporal coverages and are relatively inexpensive. However, the satellite does not directly measure Ta but only the LST.

Based on the physical linkage between LST and Ta, several authors have offered methods to estimate Ta using remote sensing satellite data [5,18,19]. During past decades, a large body of research has been collected regarding the retrieval of LST from satellite-based thermal infrared (TIR) data [20], in particular that related to the better understanding of emissivity and atmospheric effects [18,21,22]. As a result, LST can be retrieved nowadays relatively accurately from remotely sensed TIR data [23]. Several studies have demonstrated that Ta and LST data are highly correlated [24]. However, as expected, large differences have been noticed [25] which are related, for example, to physical properties and atmospheric conditions [26].

Three major approaches have been used to estimate Ta from LST data [27]:

(1) energy-balance parameterization based on thermodynamic approaches [15,19,28],
(2) contextual approaches based on temperature–vegetation index relations (TVX) [5,29,30], and
(3) statistical approaches using various form of regression techniques [15,16,23,31–34].

Good exemplary studies estimating air temperatures with MODIS LST products using the aforementioned methods can be found for example in Bartkowiak et al. [35], Lu et al. [36], Zhou et al. [37], Janatian et al. [38], Ho et al. [39], Duan et al. [40], and Benali et al. [3].

Within the last two decades, statistical approaches, including simple and advanced regression, (e.g., linear and multiple regression, and machine learning techniques) have been developed to estimate Ta from Moderate Resolution Imaging Spectroradiometer (MODIS) LST products with varying levels of success. More recently, several studies have investigated more complex and advanced approaches to estimate Ta from MODIS LST products, such as geographically weighted regression (GWR) and climate space weighted regression (CSWR) [6], spatiotemporal regression-kriging (STRK) [31], stepwise [38,41], random forest (RF) [39,41–45], generalized boosted model (GBM) [45], cubist [41,45], support vector machine (SVM) [39], ordinary least squares (OLS) [39] and M5 model tree [46]. To ensure high modeling accuracy, several papers have highlighted the usefulness of multivariate and non-parametric algorithms such as RF and STRK. For instance, Kilibarda et al. [31] estimated mean, maximum, and minimum daily Ta with a spatial resolution of 1 km at a global scale using STRK with MODIS 8-day time-series LST products along with elevation, wetness index, and geographical location. The performance of STRK to predict Ta from MODIS LST products was compared with the performance of the linear regression model. The results indicated that the root-mean-square errors (RMSEs) for predicting mean, maximum, and minimum daily Ta

are ±2 °C for areas with a high density of stations and from ±2 °C to ±4 °C for areas with a coarse station density. The lowest accuracy was 6 °C in Antarctica and at locations with high altitudes. Yoo et al. (2018) [43] estimated maximum and minimum daily Ta in two megacities using LST data from MODIS Terra/Aqua and seven auxiliary variables based on the RF machine learning method, resulting in an RMSE of 1.1 °C and 1.2 °C for maximum and minimum Ta, respectively, in Seoul, and an RMSE of 1.7 °C and 1.2 °C for maximum and minimum Ta, respectively, in Los Angeles. Several authors have concluded that machine learning techniques perform better than more conventional methods which provide multi-variables and nonlinear and nonparametric regression and classification [38,39,43,45,47,48]. Machine learning algorithms are particularly useful for cases where no deterministic model is available to solve the problem. Our research objective was to develop a robust empirical model to estimate climatologies of average monthly Ta across Mongolia at 1 km spatial resolution using time-series of MODIS Terra LST products, terrain parameters (elevation, slope, and aspect), and other ancillary information.

## 2. Study Area

The area studied in this work covers the entirety of Mongolia with a total area of approximately $1566 \times 10^6$ km². Mongolia is the eighteenth largest and most sparsely populated country in the world. Mongolia extends between the latitudes 41°35′ N–52°09′ N and the longitudes 87°44′ E–119°56′ E, with an average land surface elevation of 1580 m above sea level [49]. The elevation ranges between 524 m to 4320 m above sea level (Figure 1a). Its continentality increases from east to west. The climate conditions are extreme continental with semiarid and arid regions (Figure 1b). The "blue sky" country counts on average 260 sunny days per year and is characterized by a long cold winter and a short dry-hot summer with generally low precipitation [50]. The country's average annual temperature varies between –8 °C and 6 °C [49] with strong temperature gradients. Annual total precipitation ranges between 50 mm in the desert steppe and desert regions to 500 mm in the high mountain regions. Generally, precipitation gradually increases from south to north.

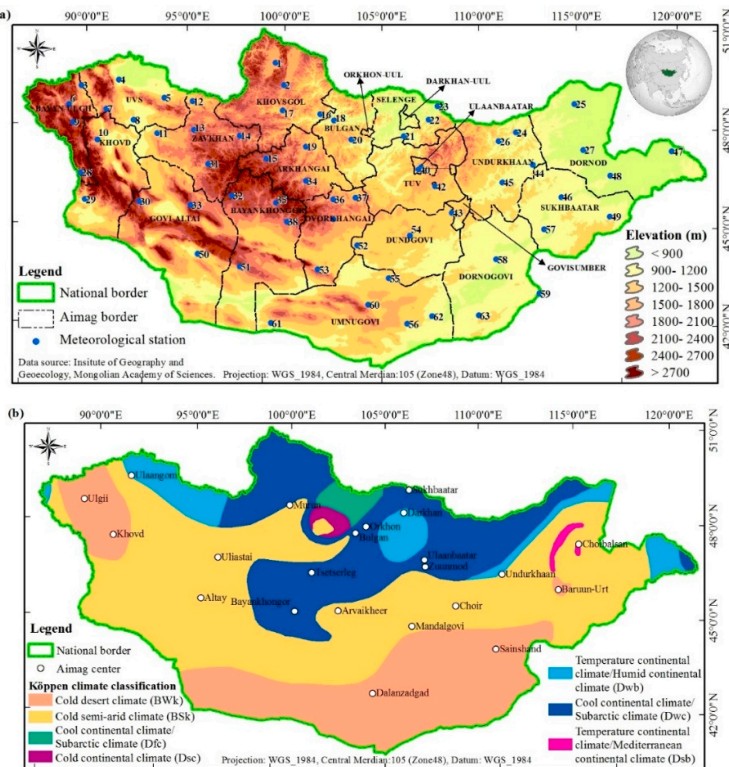

**Figure 1.** Mongolia: (**a**) Topography and location of meteorological stations (*n* = 63) for which reference air temperature information from automatic weather stations was available. The digital

elevation model (DEM) was derived from the Shuttle Radar Topography Mission (SRTM-DEM) with a resolution of 90 m. (**b**) Köppen climate classification of Mongolia based on long-term meteorological conditions (>30 years) [51].

According to the Köppen climate classification, Mongolia can be divided into seven climatic regions (Figure 1b). The most prevalent climates are BWk, BSk, and Dwc. By contrast, Dfc, Dsc, Dwb, and Dsb are less prevalent. Exemplary temperature and precipitation charts for the three most prevalent climates are shown in Figure 2, demonstrating a large variation in precipitation and temperature variation.

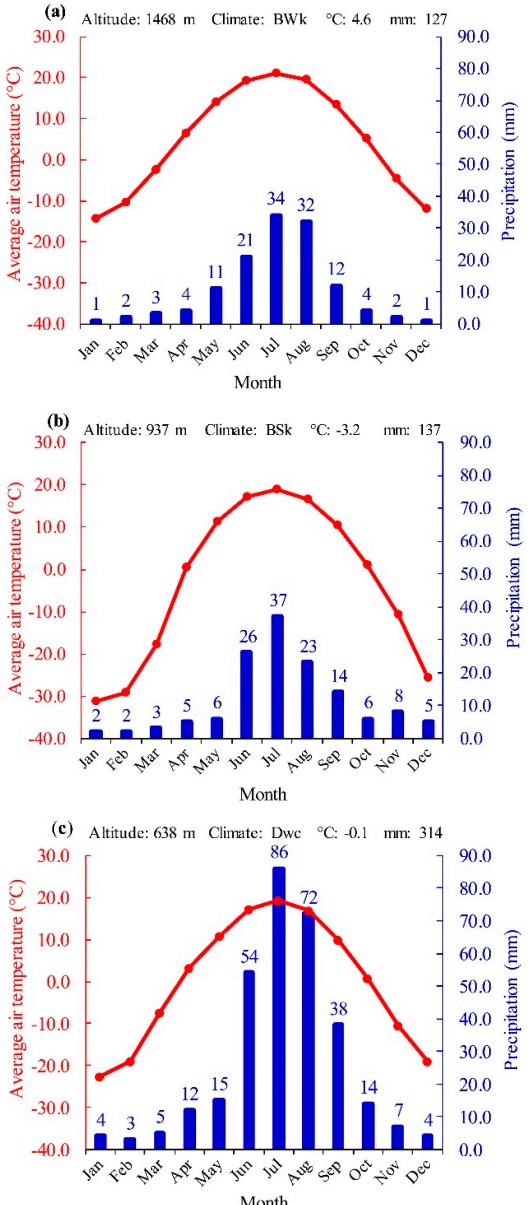

**Figure 2.** Climate graphs of the three most prevalent climatic zones in Mongolia [52]. (**a**) Dalanzadgad with cold desert climate (BWk), (**b**) Ulaangom with cold semi-arid climate (BSk), and (**c**) Sukhbaatar with cool continental climate/subarctic climate (Dwc). Information about altitude, average annual air temperature, and total precipitation is provided in each sub-plot. The data reflect long-term averages (>30 years).

## 3. Data and Methods

### 3.1. Remote Sensing Data

MODIS LST products are distributed by the Land Processes Distributed Active Archive Center (LP DAAC) in a hierarchical data format or HDF file. We used observations from MOD11 from the Terra satellite. MODIS generates two daily observations which involve one for daytime (LSTd) and one for nighttime (LSTn) at approximately 10:30 and 22:30 local time, respectively. The new collection 6 (c006) of MODIS LST products have been used to estimate Ta [53]. This dataset was made available in 2016. It covers the entire period (2002–2017) and data are of higher quality compared to the earlier collection(s), which had been used for previous studies such as [3,17,46,47,54,55]. The LST accuracy of the c006 products is reported as being approximately twice as good as collection 5 (c005) due to the incorporation of the emissivity adjustment model in the MODIS split-window algorithm [53]. For instance, the c006 LST product reduced the RMSE of bare soil sites of the c005 LST product by 1.24 °C during the day and 0.58 °C at night ([53], p. 88).

To cover the land surface of Mongolia, seven tiles of granules with horizontal (h) and vertical (v) title numbers h23v03, h23v04, h24v03, h24v04, h25v03, h25v04, and h26v04 had been used. The MODIS MOD11A2 c006 data were obtained through the online Data Pool at the National Aeronautics and Space Administration (NASA), the LP DAAC, and the United States Geological Survey (USGS) Earth Resources Observation and Science (EROS) Center, Sioux Falls, South Dakota. The retrieved MODIS LST used the generalized split-window algorithm [56] to derive surface temperature from the recorded at-satellite radiances.

For this study, we used the MODIS Terra 8-day LST product (MOD11A2) at a spatial resolution of 1 km, gridded in the Sinusoidal projection intervals, and covering the period 2002–2017. The HDF file for the MOD11A2 product includes 12 different scientific data sets (SDSs), as shown in Table 1. A detailed description of SDSs is given in [57,58]. This MOD11A2 product includes daytime and nighttime LST data (LSTd and LSTn), quality information (QCd and QCn), observation information (DvA, NvA, DvT, and NvT), emissivity data (Em31 and Em32), and clear sky coverage (CsD and CsN). The HDF file for this product also contains associated quality science dataset layers which provide users with information regarding the usability and usefulness of the data products. The MODIS LST quality science dataset layers are binary encoded and bit packed. The quality assurance (QA) layer containing integer values had been converted to a bit binary value for interpretation [59,60] (p. 19). The quality controls (QC) are defined by bit flags such as mandatory quality assessment (QA) flags, data quality flags, emissivity quality flags, and cloud error flags.

To retrieve and pre-process the products, the *MODIS* R-package (MODIS acquisition and processing package v1.1.4) was used [61]. The package is run in the R software system and environment for statistical computing and graphics [62]. The *MODIS* R-package allows automatic downloading of data and processing such as changing file format, mosaicking, subsetting, and time-series filtering [63]. Using the package, digital numbers (DN) of MODIS Terra LST products were converted into LST (Table 1). Additionally, three terrain parameters (elevation, slope, and aspect) originating from SRTM-DEM [64] were retrieved. All raster data were re-projected to MODIS sinusoidal projection.

**Table 1.** Description of Moderate Resolution Imaging Spectroradiometer (MODIS) land surface temperature (LST) products used in this study (source: Land Processes Distributed Active Archive Center (LP DAAC), 2019).

| Variable Type | Acronym | Units | Data Type | Fill Value | Valid Range (VR) | Scale Factor (SF) | Additional Offset (AO) |
|---|---|---|---|---|---|---|---|
| Daytime LST | LSTd | Kelvin | 16 bit | 0 | 7500 to 65,535 | 0.02 | N/A |
| Nighttime LST | LSTn | Kelvin | 16 bit | 0 | 7500 to 65,535 | 0.02 | N/A |
| Day clear sky coverage | CsD | N/A | 16 bit | 0 | 1 to 65,535 | 0.0005 | N/A |
| Night clear sky coverage | CsN | N/A | 16 bit | 0 | 1 to 65,535 | 0.0005 | N/A |
| View zenith angle of daytime | DvA | Degree | 8 bit | 255 | 0 to 130 | 1.0 | −65 |
| View zenith angle of nighttime | NvA | Degree | 8 bit | 255 | 0 to 130 | 1.0 | −65 |

| Time of daytime (local solar) | DvT | Hours | 8 bit | 255 | 0 to 240 | 0.1 | N/A |
| Time of nighttime (local solar) | NvT | Hours | 8 bit | 255 | 0 to 240 | 0.1 | N/A |
| Emissivity band 31 | Em31 | None | 8 bit | 0 | 1 to 255 | 0.002 | 0.49 |
| Emissivity band 32 | Em32 | None | 8 bit | 0 | 1 to 255 | 0.002 | 0.49 |
| Quality controls of day LST | QCd | Bit | 8 bit | N/A | 0 to 255 | N/A | N/A |
| Quality controls of night LST | QCn | Bit | 8 bit | N/A | 0 to 255 | N/A | N/A |

Note: Digital number (DN) = VR × SF − AO.

### 3.2. In Situ Meteorological Data

Sixty-three synoptic weather stations are present in Mongolia. Their geographical locations are indicated in Figure 1a. The weather stations provide Ta every 3 h, i.e., eight times a day. Air temperature data between 2002 and 2004 were obtained from the Mongolian Information Research Institute of Meteorology, Hydrology, and the Environment (IRIMHE). Data from 2004 to 2017 was downloaded from the "Reliable Prognosis (RP5)" website (https://rp5.ru). From the three-hourly meteorological data, the average air temperature was calculated for every 8 days of MODIS LST, taking into account the eight daily observations. This led to a total of 8544 meteorological data-points from 63 automatic weather stations covering the same period as the MODIS LST products, allowing for the development of prediction models between the remotely sensed data and Ta. The frequency distribution of the measured Ta reference data from 63 weather stations for the period 2002–2017 is shown in Figure 3 (*n* = 8544). The value of measured Ta ranged from –36.6 °C to 27.2 °C with a mean value of 0.7 °C and a standard deviation of 14.6 °C.

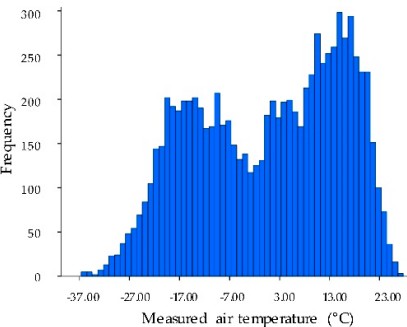

**Figure 3.** Frequency distribution of measured average 8-day air temperature reference data (*n* = 8544) from the 63 automatic weather stations for the years 2002–2017. Monthly statistics are depicted in Table 2.

### 3.3. Random Forest and Partial Least Square Regression

RF and PLS models were trained to predict Ta using up to 17 predictor variables. The use of two competing approaches permits the evaluation of the benefits of using non-linear machine learning approaches (e.g., RF) compared to classical linear regression models (e.g., PLS). Twelve of the seventeen variables were derived from LST time-series products of the Terra MODIS for the period 2002–2017 (Table 1). The five remaining variables were elevation, slope, and aspect (extracted from SRTM-DEM), and geographical location (latitude and longitude) of weather stations (extracted from vector data). Summary descriptive statistics of the response and the 17 predictor variables are reported in Table 2. Longitude was included as this indirectly depicts (for Mongolia) distance to sea [65].

**Table 2.** List of response/predictor variables and corresponding descriptive statistics (period 2002 to 2017). The list includes the measured air temperature (Ta) reference data at the weather station level ($n$ = 712 for each of the twelve months) as well as the corresponding seventeen predictor variables extracted from satellite and other geo-data. For the acronyms of the variables, see Table 1.

| Variable | No. of Samples ($n$) | Minimum | Maximum | Mean | Standard Deviation |
|---|---|---|---|---|---|
| $Ta_{01}$ | 712 | −36.60 | −6.60 | −20.83 | 5.36 |
| $Ta_{02}$ | 712 | −35.10 | −0.60 | −16.60 | 5.38 |
| $Ta_{03}$ | 712 | −20.50 | 4.80 | −6.73 | 4.22 |
| $Ta_{04}$ | 712 | −7.60 | 12.40 | 3.92 | 3.32 |
| $Ta_{05}$ | 712 | 2.80 | 19.30 | 10.75 | 3.05 |
| $Ta_{06}$ | 712 | 9.20 | 24.90 | 16.98 | 3.06 |
| $Ta_{07}$ | 712 | 11.60 | 27.20 | 19.49 | 3.21 |
| $Ta_{08}$ | 712 | 8.70 | 25.60 | 17.16 | 3.34 |
| $Ta_{09}$ | 712 | 2.60 | 19.70 | 10.48 | 3.07 |
| $Ta_{10}$ | 712 | −8.20 | 11.70 | 1.49 | 3.08 |
| $Ta_{11}$ | 712 | −22.70 | 0.30 | −9.62 | 3.76 |
| $Ta_{12}$ | 712 | −31.50 | −6.10 | −17.85 | 4.40 |
| LSTd | 8544 | −36.90 | 48.60 | 13.43 | 20.08 |
| LSTn | 8544 | −42.50 | 24.40 | −5.75 | 14.47 |
| CsD | 8544 | 0.00 | 0.13 | 0.06 | 0.02 |
| CsN | 8544 | 0.00 | 0.13 | 0.07 | 0.02 |
| DvA | 8544 | −55.00 | 62.00 | 5.04 | −52.35 |
| DvT | 8544 | 10.40 | 12.10 | 11.82 | 0.71 |
| Em31 | 8544 | 0.96 | 0.99 | 0.98 | 0.50 |
| Em32 | 8544 | 0.97 | 0.99 | 0.98 | 0.49 |
| NvA | 8544 | −65.00 | 56.00 | −0.48 | −56.47 |
| NvT | 8544 | 20.80 | 22.70 | 21.90 | 1.29 |
| QCd | 8544 | 2.00 | 133.00 | 61.98 | 16.58 |
| QCn | 8544 | 2.00 | 145.00 | 55.14 | 19.70 |
| Elevation | 63 | 667.00 | 2255.00 | 1369.10 | 411.70 |
| Slope | 63 | 0.08 | 19.60 | N/A | N/A |
| Aspect | 63 | 6.34 | 358.10 | N/A | N/A |
| Latitude | 63 | 42.97 | 51.11 | N/A | N/A |
| Longitude | 63 | 89.93 | 118.67 | N/A | N/A |

The correlation matrix (Figure 4) reveals a strong correlation between Ta and daytime/nighttime LST of MODIS, as well as several other correlations and redundancies. Based on these intercorrelations and taking into account that the number of variable sets should be relatively small, the predictor variables were grouped into seven different groupings (Table 3).

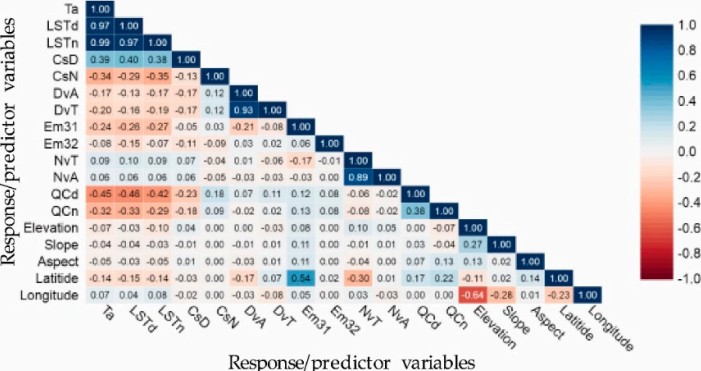

**Figure 4.** Correlation matrix between response and predictor variables ($n$ = 8544). The saturation of the colors indicates the strength of the correlations. Positive correlations are shown in blue and negative correlations in red. In this graph, the air temperature data has been pooled across the twelve months. For the abbreviations, see Table 1.

**Table 3.** Seven model subsets studied. The seven groups were generated to study the relations between responses and up to 17 predictor variables. $N_{var}$ indicates the number of predictor variables in each group.

|  | Acronym | Variables | $N_{var}$ |
|---|---|---|---|
| Group 1 | G1 | LSTd and LSTn | 2 |
| Group 2 | G2 | LSTd, LSTn, and elevation | 3 |
| Group 3 | G3 | Elevation, slope, aspect, latitude, and longitude | 5 |
| Group 4 | G4 | Combined G1 and G3 | 7 |
| Group 5 | G5 | CsD, CsN, DvA, DvT, Em31, Em32, NvA, NvT, QCd, and QCn | 10 |
| Group 6 | G6 | Combined G1 and G5 | 12 |
| Group 7 | G7 | Combined G1, G3, and G5 | 17 |

3.3.1. RF Regression

The well-known random forest regression method [42,44] was chosen as the main approach to model the relation between our response variable (Ta) and the predictor variables listed in Tables 2 and 3 (LST MODIS products plus elevation, slope, aspect, latitude, and longitude). RF is a non-linear statistical ensemble method that leverages uncorrelated decision trees for regression. Developed by Breiman [66], it is capable of modeling discrete and/or continuous data sets [67,68]. RF predictions are obtained by aggregating a large number of individual regression decision trees where each decision tree is built from bootstrapped training samples (as in bagging) and variables are randomly selected at each decision node. The algorithm then randomly selects a subset of the predictors as candidates for splitting [66,69]. To obtain the final regression model, the results of all the individual trees are averaged. Good examples of the benefits and drawbacks of RF are given in [70–72].

The RF algorithm provides out-of-bag error (OOB) estimates and variable importance rankings [73,74], as not all observations are included in the respective bootstraps of the individual trees. In each tree at each split, the enhancement in the split-criterion importance measure is characterized by the splitting variable and aggregates individually all the trees in the forest for each variable [75]. Variable importance is measured by computing the increase in mean square error (MSE) when the OOB data for each variable are again computed but without the left-out variable [66,76]. The variable importance measures can assist in defining which variables are most important in the reduction of prediction error [71]. Two kinds of variable importance measures widely use the "randomForest" package in R [69,77]: (1) percent increase in the mean square error (%IncMSE) and (2) increase in node purity (IncNodePurity). From these, our analysis computed and analysed %IncMSE. However, we also checked the IncNodePurity indicator, but found similar results (not shown). In our research, the basic algorithm shown in Equation (1) was used to build the RF predictor for regression [75]:

$$\hat{f}(x) = \frac{1}{B}\sum_{b=1}^{B} T_b(x) \tag{1}$$

A new bootstrap sample for each decision tree $T_b$ that includes X = $x_1$...... $x_i$ with responses Y = $y_1$...... $y_i$ bagging repeatedly (B times) selects a random sample from training data and each unpruned decision tree is increased in the sample. To increase each individual tree $T_b$, the following steps are repeated at each terminal node of the tree:

- Randomly select *m* variables from *p* variables
- Pick the variable that best splits and the corresponding split point
- Split the node into two nodes.

As mentioned above, to implement the RF regression model, two parameters must be set: the number of decision trees ($n_{tree}$) and the number of variables to select for the best split ($m_{try}$). For both hyperparameters, standard settings have been chosen. Each decision tree is independently increased to its maximum size, focusing on a new bootstrap sample from the training data (2/3 of samples). The remaining 1/3 of the samples, not used to fit the given decision tree, are referred to as the out-of-bag sample. The OOB sample is used to calculate the OOB error rate and variable importance. For quantifying the OOB error (prediction error) for each RF decision tree, we used Equation (2), i.e.,

$$Error_{OOB} = \frac{1}{n}\sum_{i=1}^{n}(y_i - \hat{y}_i)^2 \tag{2}$$

where $\hat{y}_i$ is the estimated output of OOB samples, $y_i$ is the actual output, and $n$ is the total number of OOB samples. RF regression is flexible and easy to use in comparison to other machine learning algorithms, even without hyper-parameter tuning.

### 3.3.2. PLS Regression

For comparison—and to assess the differences between linear and non-linear models—a prominent linear modeling technique was used: partial least square regression. PLS is widely used by the remote sensing community for vegetation analysis [78–80], soil related studies [81–83], and climate and ecological studies [84–87] amongst others.

PLSR is a multivariate linear regression method used to predict a set of dependent variables from a set of independent variables or predictors [88]. PLSR was originally developed for econometrics and chemometrics [89], where commonly a large number of strongly correlated predictor variables exist [90]. PLSR reduces the variables to a smaller set of uncorrelated components and performs least squares regression on these components instead of on the original data. Compared to other techniques, PLSR is more robust and less susceptible to data redundancy and over-fitting [91].

PLSR extracts a set of latent variables that explain the correlation between dependent and independent variables. The optimum number of latent variables for each generated model are implemented using the minimum value of residual mean squared error and the leave-one-out-cross-validation (LOO-CV) methods, e.g., jackknife and bootstrap [92]. To assess which variables are most contributing to the PLS model, we used the variable importance in projection (VIP) method [90], as seen in Equation (3), i.e.,

$$VIP_j = \sqrt{\frac{\sum_{f=1}^{F} w_{jf}^2\, SSY_f\, J}{SSY_{total}\, F}} \tag{3}$$

where $VIP_j$ is a measure of the contribution of the $j$ variable in the PLSR model, $W_{jf}$ is the weight value for the $j$ variable and $f$ latent variables (components), $SSY_f$ is the sum of squares of explained variance for the $f$ latent variable and $J$ number of the predictor (independent) variables, $SSY_{total}$ is the total sum of squares explained as the response (dependent) variables, and $F$ is the total number of latent variables. The VIP values determine the contribution of the predictor variables to the PLSR latent variables. A VIP value greater than 0.80 ensures that only relevant variables are considered [90]. In [93] the VIP threshold of predictor variables that were identified as the most relevant variables ranged between 0.83 and 1.21. Predictor variables with ≤0.80 VIP values were classified as less important while variables with VIP values ≥1.20 were considered the most influential.

### 3.3.3. Model Evaluation and Statistics

Two widely used statistics were calculated to assess the accuracy of the models [94], including the $R^2$ and the RMSE. The $R^2$ describes the percentage of explained variance whereas the RMSE summarizes the deviations of predictions from the one-to-one line.

As both models provide quantitative information about the importance of different variables, we also report these findings. For the RF regression model, we assessed the importance of the individual predictors in Ta estimates focused on the %IncMSE [39]. For the PLS regression model, we used the VIP method [95,90,93].

## 4. Results

*4.1. Comparison of RF and PLS Models: Variable Importance and Prediction Accuracy*

The estimated importance of the 17 predictor variables in the RF regression model is shown in Figure 5 for each of the twelve months. Under the top three in each month, LSTn appeared 12 times, LSTd 11 times, elevation 7 times, latitude 5 times and aspect once. The remaining variables were never found under the top three in those rankings. Strong seasonality in the ranking can also be observed. For example, the warm season Ta (April to October) was strongly dependent on elevation. Conversely, the cold season Ta was more heavily affected by latitude. Using all 17 predictor variables led to $R^2$ values in the range from 0.83 (April) to 0.96 (August), while RMSE were between 0.91 °C (September) to 1.92 °C (February) (see Table 4, column G7).

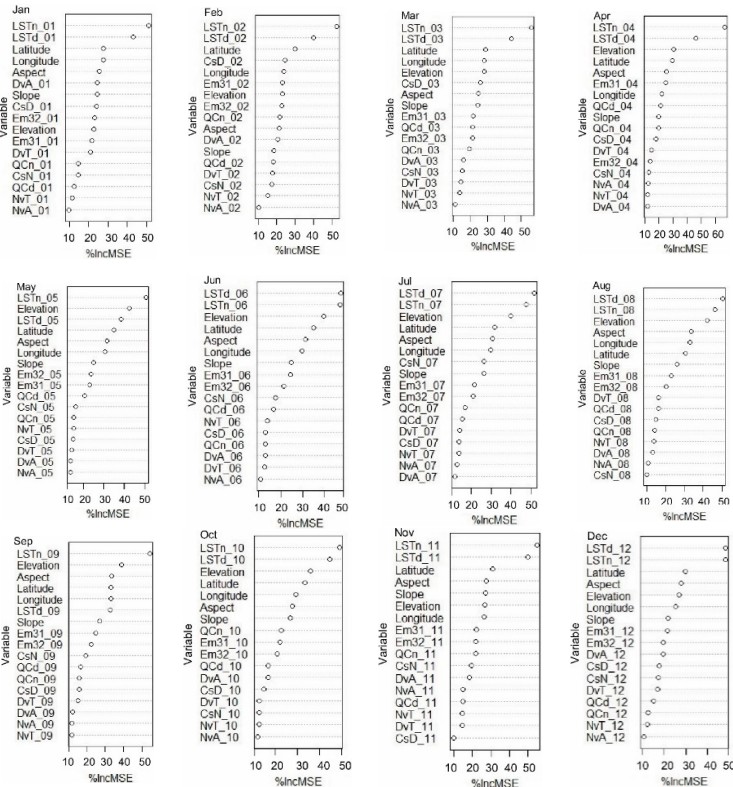

**Figure 5.** Random forest (RF) variable importance for each month. The importance is here given as the percentage increase in mean square error (%IncMSE).

Similar results were obtained for the PLS models (Appendix A Figure A1). Using PLS regression, the variables most often listed under the top three were LSTn (12 times), LSTd (12 times), elevation (4 times), latitude (4 times), and emissivity (4 times). Again, the ranking was season-dependent. The variables LSTn, LSTd, and latitude were the most important variables for estimating Ta in autumn and winter (September to February). The Ta for spring (March–May) was strongly dependent on LSTn, LSTd, and emissivity. For the summer months (June–August), LSTn, LSTd, and elevation were strongly influenced by the estimation of Ta for summer.

Using the entire set of 17 predictor variables for estimating the monthly average air temperatures, accuracies of PLS models were constantly lower compared to the RF models. The PLS models gave $R^2$ a measured and estimated monthly Ta between 0.74 and 0.86 and RMSE from 1.20 °C to 2.19 °C (Appendix A Table A2, column G7).

Concerning variable importance, the three variables LSTn, LSTd, and elevation were identical for PLS and RF regression models. This shows that LSTn, LSTd, and elevation play a key role in modeling Ta, with all other variables having a significantly smaller impact.

To further study the impact of the different predictor variables, the seven variable groupings highlighted in Table 3 were analyzed in more detail. Results for each month and the annual average air temperature are shown in Table 4 for the RF models. Compared to the full set of 17 variables

(column G7), the reduced set with only three predictor variables LSTn, LSTd, and elevation (G2) achieved comparable results, again highlighting and confirming the importance of these three predictor variables. None of the other five groupings (G1 and G3 to G5) were able to yield similar model performances. The same findings also hold for the PLS models (Appendix A Table A2 for details) but with constantly lower accuracies compared to the RF models. PLS models and groupings G1 to G7 were, therefore, skipped for the remainder of the study.

**Table 4.** Modeling results obtained using the RF regression. Reported are the monthly summary statistics (coefficient of determination ($R^2$) and root-mean-square error (RMSE)) for Ta prediction models for each of the seven groups of variables. For details of groupings G1 to G7, see Table 3.

| | *n* | G1 | | G2 | | G3 | | G4 | | G5 | | G6 | | G7 | |
|---|---|---|---|---|---|---|---|---|---|---|---|---|---|---|---|
| | | $R^2$ | RMSE | $R^2$ | RMSE | $R^2$ | RMSE | $R^2$ | RMSE | $R^2$ | RMSE | $R^2$ | RMSE | $R^2$ | RMSE |
| January | 712 | 0.85 | 2.07 | 0.91 | 1.59 | 0.52 | 3.71 | 0.89 | 1.77 | 0.40 | 4.14 | 0.88 | 1.86 | 0.90 | 1.71 |
| February | 712 | 0.83 | 2.24 | 0.87 | 1.93 | 0.49 | 3.83 | 0.86 | 2.05 | 0.46 | 3.95 | 0.85 | 2.06 | 0.88 | 1.92 |
| March | 712 | 0.76 | 2.09 | 0.85 | 1.64 | 0.55 | 2.84 | 0.83 | 1.77 | 0.46 | 3.12 | 0.81 | 1.83 | 0.85 | 1.67 |
| April | 712 | 0.77 | 1.59 | 0.82 | 1.41 | 0.47 | 2.42 | 0.80 | 1.47 | 0.38 | 2.62 | 0.79 | 1.51 | 0.83 | 1.40 |
| May | 712 | 0.76 | 1.48 | 0.88 | 1.05 | 0.75 | 1.52 | 0.84 | 1.07 | 0.39 | 2.37 | 0.81 | 1.34 | 0.88 | 1.08 |
| June | 712 | 0.77 | 1.45 | 0.88 | 1.04 | 0.76 | 1.50 | 0.88 | 1.07 | 0.31 | 2.53 | 0.81 | 1.33 | 0.88 | 1.07 |
| July | 712 | 0.84 | 1.29 | 0.93 | 0.84 | 0.81 | 1.41 | 0.91 | 0.97 | 0.32 | 2.65 | 0.87 | 1.17 | 0.87 | 0.93 |
| August | 712 | 0.83 | 1.38 | 0.95 | 0.92 | 0.83 | 1.37 | 0.90 | 0.97 | 0.35 | 2.68 | 0.86 | 1.27 | 0.96 | 0.92 |
| September | 712 | 0.81 | 1.32 | 0.91 | 0.91 | 0.84 | 1.22 | 0.88 | 1.05 | 0.40 | 2.37 | 0.85 | 1.19 | 0.91 | 0.91 |
| October | 712 | 0.83 | 1.28 | 0.90 | 0.99 | 0.70 | 1.68 | 0.88 | 1.08 | 0.47 | 2.23 | 0.85 | 1.22 | 0.89 | 1.03 |
| November | 712 | 0.81 | 1.65 | 0.87 | 1.34 | 0.54 | 2.55 | 0.85 | 1.47 | 0.42 | 2.86 | 0.84 | 1.52 | 0.86 | 1.39 |
| December | 712 | 0.84 | 1.76 | 0.89 | 1.44 | 0.60 | 2.79 | 0.88 | 1.55 | 0.40 | 3.43 | 0.86 | 1.64 | 0.89 | 1.49 |

*4.2. Maps of Predicted Air Temperatures Using RF Models with the Reduced Feature Set*

Both the results of the variable importance rankings (Figure 5) and the grouping of variables (Table 4) indicate that relatively simple RF prediction models can be built to estimate Ta using only daytime/nighttime LST and elevation information. Scatterplots between measured and estimated monthly average air temperatures using only these three predictor variables are shown in Figure 6. Corresponding maps of modeled air temperatures at 1 km spatial resolution and covering the entire land-mass of Mongolia are shown in Figure 7a (see Figure A2 and Table A3 in the Appendix A for corresponding scatterplots and maps generated using PLS models).

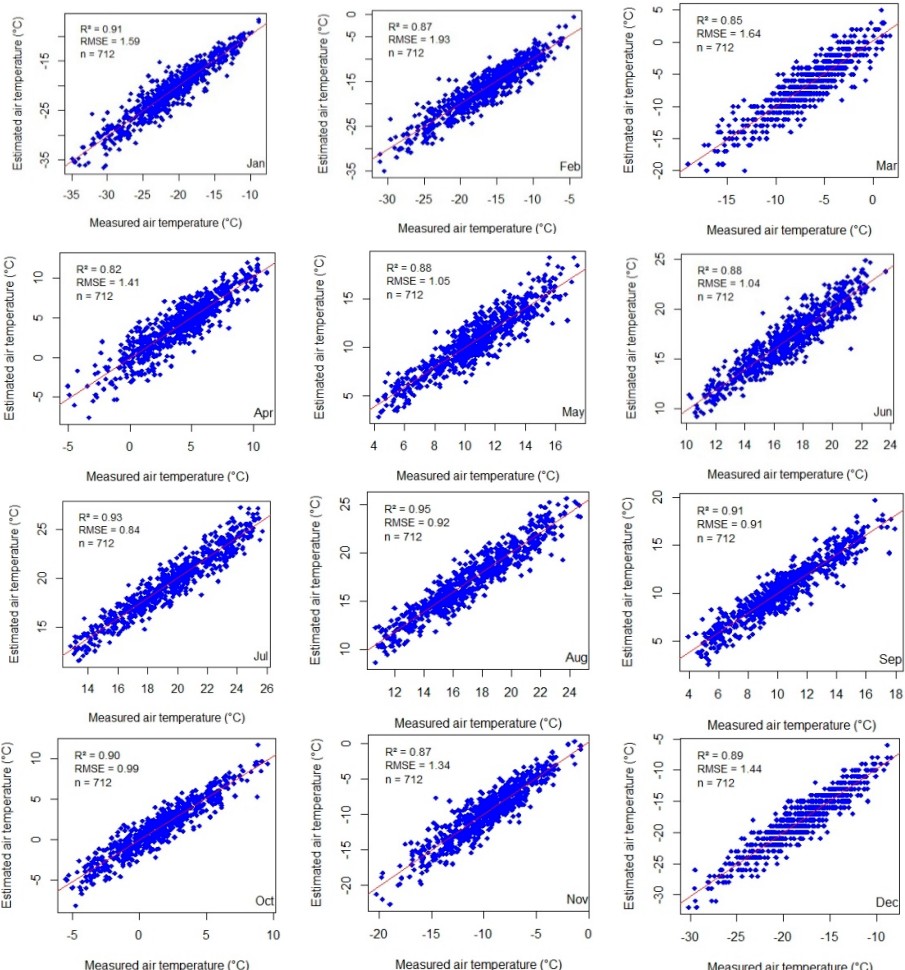

**Figure 6.** Comparison between measured and estimated monthly average Ta using LSTd, LSTn, and elevation for the RF regression model.

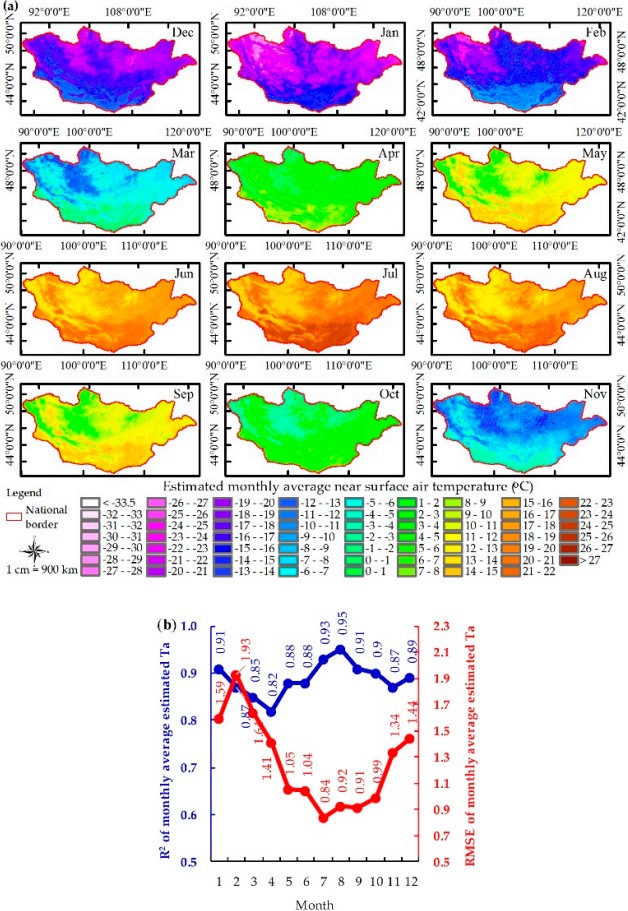

**Figure 7.** Estimated monthly average Ta based on RF regression model using LSTd, LSTn, and elevation as predictor variables. (**a**) Spatial maps of estimated monthly average Ta over Mongolia at 1 km spatial resolution. (**b**) Monthly statistics of $R^2$ (blue) and RMSE (red) between observed and predicted air temperature.

The scatterplots in Figure 6 reveal that the RF-predicted Ta is well distributed around the 1-to-1 line, with no apparent systematic deviations. In particular, we do not see any autocorrelation in the errors, nor saturation effects. The errors are generally low and the explained variance ($R^2$) mostly above 0.85. Generally, however, the RMSE increases slightly during the colder months (Figure 7b).

The maps in Figure 7a depict in high spatial detail the model predictions. As expected, the predicted air temperatures decrease with elevation (Figure 1) but reveal additional detail and information. Monthly analyses of the coefficient of determination ($R^2$; in blue) and root mean square error (RMSE; in red) for the period 2002–2017 are shown in Figure 7b. Overall, a good agreement between observed and estimated Ta values was found but reflected again the afore-mentioned seasonal pattern. Large discrepancies were found to occur in transition months, such as the start or end of seasons.

The PLS-generated maps of monthly average air temperatures are generally similar to the maps derived from RF models (see Appendix A Figure A2b). However, a more detailed analysis reveals sometimes larger differences, even if modeled air temperatures are averaged by season. For example, Figure 8 clearly shows that large method-specific differences occur (maps in the third column). The differences show large seasonal fluctuations. The deviations moreover show a clear north–south gradient with generally lower Ta estimated using PLS compared to RF (reddish colors). The deviations are usually strongest during the warmer months. As the RF model outperformed the PLS model when evaluated against the observed air temperatures (Table 4 for RF and Appendix A Table

A2 for PLS), we interpret these findings as mainly being the result of a systematic underestimation of Ta by the PLS model.

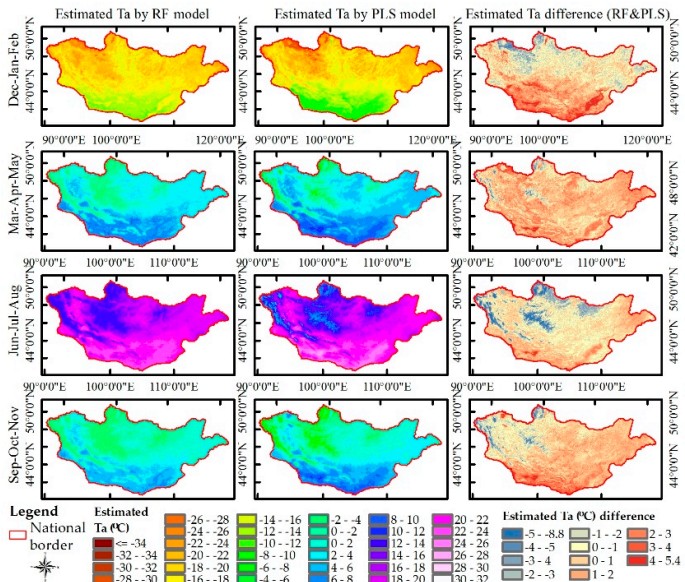

**Figure 8.** Estimated average Ta per season using LSTd, LSTn, and elevation as predictor variables. (**a**) Spatial maps of seasonal-average Ta over Mongolia at 1 km spatial resolution using the RF (first column) and partial least squares (PLS) regression models (second column). In the last column, the difference between the two model outputs is shown.

## 5. Discussion

Using relatively simple RF models driven by a few predictor variables, climatologies of monthly air temperatures in Mongolia could be obtained in this study with high accuracy (RMSE of about 0.84–1.93°C). Without any hyperparameter tuning, the non-linear RF models outperformed linear PLS models in accordance with other studies [43,55,96,97]. Amongst the variables studied, the MODIS-derived land surface temperatures (day and night) together with elevation were the three most important predictors. The studies of Noi et al. [41] and Kilibarda et al. [31] have also reported the high importance of day- and nighttime LST observations as well as elevation.

As LST (both day and night) and elevation can be readily produced at 1 km spatial resolution, the models calibrated against weather station data permitted the creation of maps of average air temperature for each of the twelve months in unprecedented detail and accuracy. Although the RF-generated maps often follow elevation, the inclusion of remotely sensed land surface temperature from MODIS improved the accuracy and spatial detail.

The results of the importance analysis indicated that nighttime LST was slightly more important compared to LSTd. The same result has been noted in China [42] and in Portugal [3]. In accordance with these studies, we argue that nighttime observations are probably more predictive because LSTn is not affected by reflected solar radiation when using TIR sensors [17]. The daytime land surface temperature was nevertheless found to be important, as it reveals the strength of the latent heat flux and the energy available for generating sensible heat [42].

Other studies have confirmed that Ta predictions are possible using satellite observations and that there is a strong relationship between Ta, LSTn, and LSTd [32]. Several studies have produced Ta estimations using MODIS LST data using multivariate linear and non-parametric regression methods [39,41-45,93]. These already published studies have showed different levels of success. The performance of multivariate and non-parametric regression models has been strongly dependent on environmental parameters such as vegetation cover, slope, aspect, elevation, quality of MODIS LST products, and applied filter techniques. For instance, the accuracy of the MODIS LST has been found

to depend on the employed split-window algorithm, cloud cover, and terrain parameters [32]. Nonetheless, estimation of Ta derived from MODIS LST studies using multivariate and non-parametric algorithms is suitable for generating results at high accuracy. For instance, [42] estimated monthly average Ta for the territory of China at a spatial resolution of 1 km using RF regression with MODIS LST, normalized difference vegetation index (NDVI), nighttime light and elevation. Using this dataset, the RMSE of the monthly average Ta ranged between 1.57 °C to 1.99 °C. Our study has showed that monthly average Ta can be accurately estimated using LSTn, LSTd, and elevation with similar RMSE ranging from 0.91 °C to 1.93 °C. The method is relatively easy to implement provided that there is a sufficient amount of training data with corresponding EO time-series observations [96].

## 6. Conclusions

In this study, PLS and RF regression models were applied to estimate monthly average Ta in Mongolia for the period 2002–2017 using MODIS LST time-series products and terrain parameters. Meteorological data from 63 automatic weather stations were used to calibrate and validate the PLSR and RF models**. Both models were trained to predict Ta using up to 17 variables as predictor variables. Twelve variables were derived from LST time-series products of Terra MODIS and three variables were extracted from an SRTM DEM (elevation, slope, and aspect). The geographical location (longitude and latitude) was used as an additional variable. For training, a total of 8544 meteorological data points from 63 automatic weather stations and corresponding MODIS LST were used. Both datasets covered the period 2002–2017. Using only day/nighttime LST and elevation as predictor variables, the correlation between measured and estimated monthly average Ta RMSE ranged from 1.20 °C to 2.19 °C for the PLSR and 0.84 °C to 1.93 °C for the RF. The significantly lower errors of the RF models confirm the benefits of this machine learning approach compared to traditional (linear) modeling techniques (e.g., PLSR). We therefore recommend the use of RF models for similar studies.

Concerning the MODIS land surface temperature data, we found that this information contributed significantly to the modeling of air temperature. For example, it was not possible to obtain similarly low errors in the modeled air temperature using only terrain parameters as predictors. It is recommended that day- and nighttime LST be used simultaneously as both variables scored high in the feature importance metric.

Both machine learning models (RF and PLSR) represented well seasonal and spatial variations in Ta when time-series of LST were included as predictor variables. Using the models, maps of the monthly average Ta of Mongolia were developed at a spatial resolution of 1 km which were representative for the period 2002–2017. Although errors in the predicted Ta were generally low, the residual errors showed a significant seasonality; the warmer months were generally better modeled compared to the extremely cold winter months. Probably, the increased errors during the winter months reflect a lower accuracy in the input (LST) data. Further research is warranted to better understand the seasonality of the model quality.

Despite these trends, we firmly believe that this spatial dataset may be useful for various environmental applications; for instance, it may be useful for better assessing bioclimatic variations within the huge land-mass of Mongolia. The developed methodology is relatively easy and transparent and can be applied in different geographic regions, provided that enough weather stations are available to permit a model calibration. The spatial resolution of the final map product mainly depends on the ground sampling distance of the employed satellite sensors. As sensor technology advances at a rapid pace, the current 1 km spatial resolution can be further improved in the near future.

**Author Contributions:** The first author M.O. analyzed the data and performed the experiments; she also runs the statistical analysis in R software and drafted the first version of the manuscript. The second author C.A. contributed to data analysis and the revision on an earlier and on the final version of the manuscript. The third author M.M. contributed MODIS data analysis preprocessing in R software. All authors together developed and discussed the manuscript and wrote the paper.

**Funding:** This article was funded by the Eurasia-Pacific Uninet, Ernst Mach Grant (reference no. ICM-2018-10238).

**Acknowledgments:** We are grateful to Anja Klisch and Valentin Pesendorfer, who are researchers at the Natural Resources and Life Sciences University (BOKU), for their assistance in writing R code. Munkhdulam Otgonbayar received funding from Eurasia-Pacific Uninet on behalf of the Austrian Federal Ministry of Science, Research and Economy (BMWFW) during the preparation of the manuscript. We thank all colleagues at the Institute of Geography and Geoecology, Mongolian Academy of Sciences who helped in this study. We would like to thank our reviewers for providing valuable feedback and suggestions.

**Conflicts of Interest:** The authors declare no conflict of interest.

## Appendix A

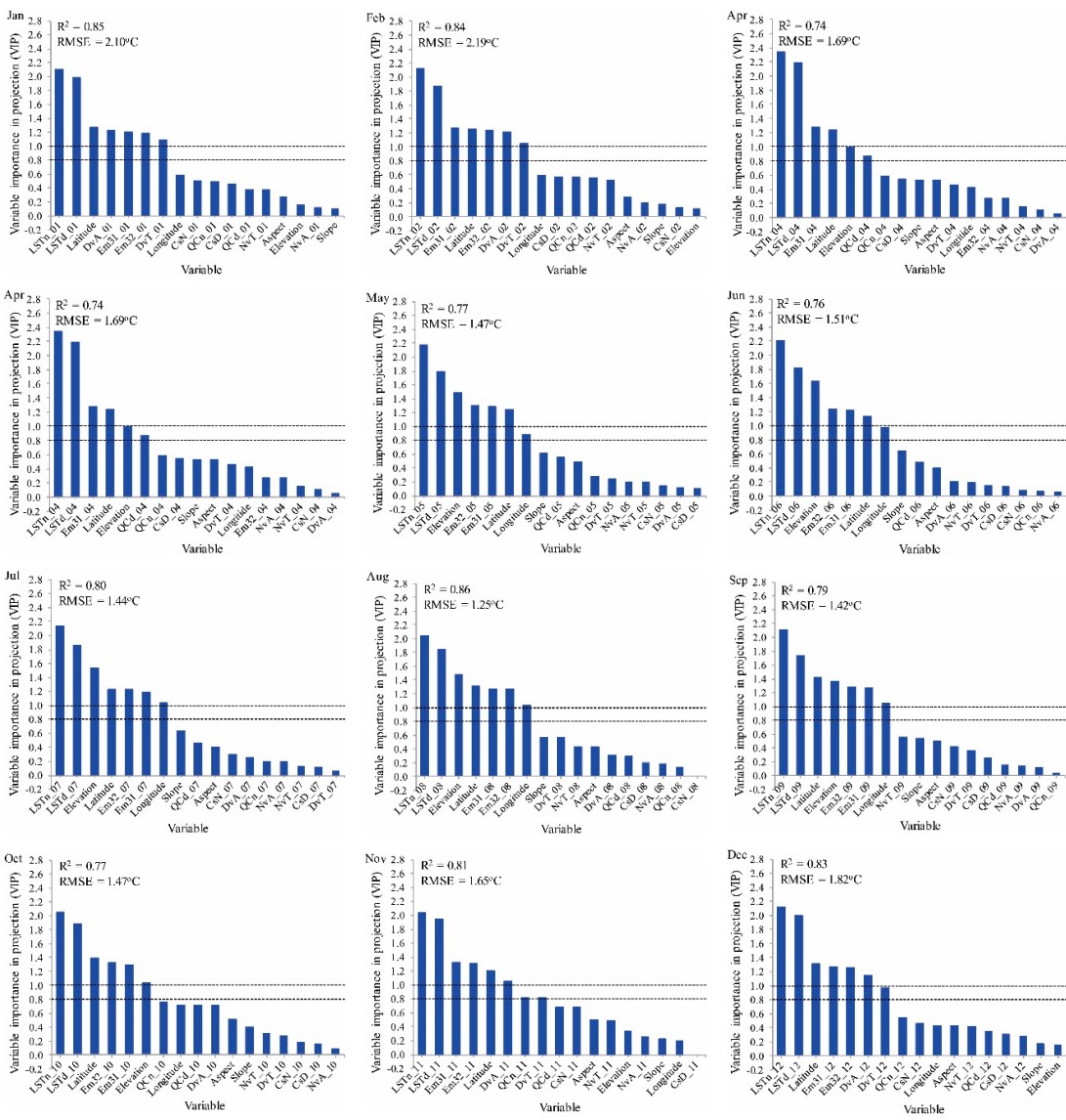

**Figure A1.** Variable importance in projections (VIPs), $R^2$, and RMSE for the 17 predictor variables in the twelve-monthly PLSR models.

**Table A1.** Model coefficients for the twelve-monthly PLS regression models including all 17 predictor variables. In the last column, the model of the annual average Ta is also included.

| Variable | Jan | Feb | Mar | Apr | May | Jun | Jul | Aug | Sep | Oct | Nov | Dec | Year |
|---|---|---|---|---|---|---|---|---|---|---|---|---|---|
| Intercept | -52.83 | -93.62 | 26.17 | 55.52 | 93.68 | 89.37 | 97.27 | 45.11 | 68.76 | 62.92 | 53.54 | 22.17 | 82.62 |
| LSTd | 0.255 | 0.219 | 0.208 | 0.185 | 0.159 | 0.146 | 0.137 | 0.178 | 0.150 | 0.128 | 0.197 | 0.234 | 0.319 |
| LSTn | 0.420 | 0.457 | 0.425 | 0.265 | 0.223 | 0.232 | 0.227 | 0.284 | 0.210 | 0.196 | 0.344 | 0.407 | 0.464 |
| CsD | 0.000* | 0.000* | 0.000* | 0.000* | 0.000* | 0.000* | 0.000* | 0.000* | 0.000* | 0.000* | 0.000* | 0.000* | 0.000* |
| CsN | -0.014 | -0.003 | 0.007 | 0.000* | 0.001 | 0.001 | 0.002 | 0.000* | 0.004 | 0.000* | 0.000* | -0.011 | 0.000* |
| DvA | 0.015 | 0.015 | 0.007 | -0.002 | 0.005 | 0.008 | 0.011 | 0.021 | 0.006 | 0.022 | 0.003 | 0.004 | -0.007 |
| DvT | 0.094 | 0.109 | -0.015 | -0.130 | -0.072 | -0.041 | -0.019 | 0.003 | -0.105 | 0.061 | -0.044 | -0.021 | 0.095 |
| Em31 | 0.069 | 0.104 | 0.021 | -0.152 | -0.133 | -0.126 | -0.131 | -0.050 | -0.127 | -0.125 | -0.047 | 0.006 | -0.079 |
| Em32 | 0.101 | 0.158 | 0.003 | -0.012 | -0.208 | -0.198 | -0.209 | -0.096 | -0.199 | -0.195 | -0.087 | -0.009 | 0.007 |
| NvA | −0.019 | -0.034 | -0.034 | -0.012 | -0.009 | -0.002 | -0.007 | -0.017 | 0.007 | -0.004 | -0.017 | -0.028 | -0.038 |
| NvT | 0.031 | 0.100 | -0.114 | 0.045 | 0.053 | 0.050 | 0.034 | 0.029 | 0.169 | 0.083 | -0.030 | -0.038 | -0.413 |
| QCd | 0.000* | 0.000* | 0.000* | 0.000* | 0.000* | 0.000* | 0.000* | 0.000* | 0.000* | 0.000* | 0.000* | 0.000* | 0.000* |
| QCn | 0.000* | 0.000* | 0.000* | 0.000* | 0.000* | 0.000* | 0.000* | 0.000* | 0.000* | 0.000* | 0.000* | 0.000* | 0.000* |
| Elevation | 0.000* | 0.000* | -0.001 | -0.001 | -0.001 | -0.001 | -0.001 | -0.002 | -0.001 | -0.001 | -0.001 | 0.000* | -0.001 |
| Slope | −0.021 | 0.028 | -0.009 | -0.061 | -0.059 | -0.062 | -0.065 | -0.047 | -0.051 | -0.037 | -0.025 | -0.031 | 0.026 |
| Aspect | -0.002 | -0.001 | -0.001 | -0.002 | -0.002 | -0.001 | -0.002 | -0.001 | -0.002 | -0.002 | -0.002 | -0.003 | 0.003 |
| Latitude | −0.411 | -0.401 | -0.248 | -0.268 | -0.236 | -0.214 | -0.246 | -0.244 | -0.260 | -0.248 | -0.241 | -0.393 | 0.336 |
| Longitude | 0.099 | 0.106 | 0.039 | 0.023 | 0.042 | 0.046 | 0.052 | 0.065 | 0.048 | 0.032 | 0.016 | 0.062 | 0.012 |

* <0.0001.

**Table A2.** PLS regression results. Summary statistics ($R^2$ and RMSE) for the monthly Ta prediction models, including six groups of variables.

| | n | G1 | | G2 | | G3 | | G4 | | G5 | | G6 | | G7 | |
|---|---|---|---|---|---|---|---|---|---|---|---|---|---|---|---|
| | | $R^2$ | RMSE | $R^2$ | RMSE | $R^2$ | RMSE | $R^2$ | RMSE | $R^2$ | RMSE | $R^2$ | RMSE | $R^2$ | RMSE |
| January | 712 | 0.87 | 1.96 | 0.87 | 1.95 | 0.34 | 4.34 | 0.82 | 2.26 | 0.39 | 4.17 | 0.86 | 2.01 | 0.85 | 2.10 |
| February | 712 | 0.83 | 2.19 | 0.83 | 2.19 | 0.32 | 4.40 | 0.79 | 2.45 | 0.37 | 4.26 | 0.85 | 2.05 | 0.84 | 2.19 |
| March | 712 | 0.80 | 1.89 | 0.74 | 1.94 | 0.36 | 3.40 | 0.77 | 2.04 | 0.39 | 3.32 | 0.81 | 1.84 | 0.81 | 1.83 |
| April | 712 | 0.79 | 1.51 | 0.79 | 1.53 | 0.48 | 2.39 | 0.77 | 1.58 | 0.32 | 2.73 | 0.75 | 1.67 | 0.74 | 1.69 |
| May | 712 | 0.76 | 1.48 | 0.79 | 1.41 | 0.74 | 1.54 | 0.80 | 1.37 | 0.31 | 2.52 | 0.76 | 1.50 | 0.77 | 1.47 |
| June | 712 | 0.78 | 1.44 | 0.80 | 1.38 | 0.75 | 1.52 | 0.79 | 1.41 | 0.26 | 2.63 | 0.78 | 1.45 | 0.76 | 1.51 |
| July | 712 | 0.83 | 1.33 | 0.86 | 1.20 | 0.79 | 1.48 | 0.84 | 1.28 | 0.28 | 2.71 | 0.82 | 1.34 | 0.80 | 1.44 |
| August | 712 | 0.84 | 1.36 | 0.87 | 1.23 | 0.81 | 1.44 | 0.85 | 1.28 | 0.30 | 2.80 | 0.83 | 1.37 | 0.86 | 1.25 |
| September | 712 | 0.81 | 1.35 | 0.84 | 1.24 | 0.82 | 1.30 | 0.83 | 1.26 | 0.32 | 2.53 | 0.80 | 1.37 | 0.79 | 1.42 |
| October | 712 | 0.83 | 1.27 | 0.83 | 1.26 | 0.68 | 1.73 | 0.82 | 1.31 | 0.41 | 2.36 | 0.81 | 1.32 | 0.77 | 1.47 |
| November | 712 | 0.83 | 1.54 | 0.83 | 1.57 | 0.37 | 2.97 | 0.79 | 1.70 | 0.40 | 2.92 | 0.82 | 1.59 | 0.81 | 1.65 |
| December | 712 | 0.86 | 1.68 | 0.86 | 1.67 | 0.36 | 3.53 | 0.82 | 1.89 | 0.34 | 3.58 | 0.85 | 1.74 | 0.83 | 1.82 |

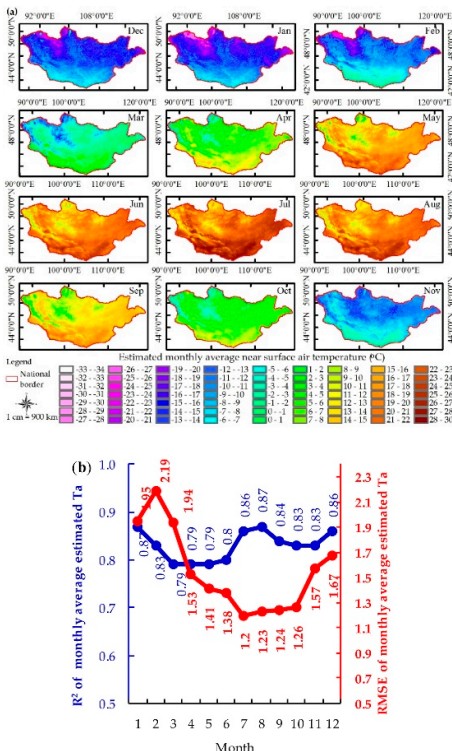

**Figure A2.** Estimated monthly average Ta based on the PLS regression model and using LSTd, LSTn, and elevation as predictor variables. (**a**) Spatial maps of estimated monthly average Ta over Mongolia at 1 km spatial resolution. (**b**) Monthly statistics of $R^2$ (blue) and RMSE (red) between observed and predicted air temperature.

**Table A3.** Model equations obtained from the PLS regression models using three variables: LSTd, LSTn, and elevation. The months are numbered from 01 to 12.

| Regression Models | $R^2$ | RMSE |
|---|---|---|
| $Ta_{01} = -2.137 + 0.347 \times LSTd + 0.497 \times LSTn + 0.00033 \times elevation$ | 0.87 | 1.95 |
| $Ta_{02} = -3.037 + 0.297 \times LSTd + 0.493 \times LSTn + 0.00019 \times elevation$ | 0.83 | 2.19 |
| $Ta_{03} = -1.986 + 0.252 \times LSTd + 0.477 \times LSTn - 0.001 \times elevation$ | 0.74 | 1.94 |
| $Ta_{04} = 0.516 + 0.296 \times LSTd + 0.424 \times LSTn - 0.002 \times elevation$ | 0.79 | 1.53 |
| $Ta_{05} = 3.863 + 0.272 \times LSTd + 0.383 \times LSTn - 0.002 \times elevation$ | 0.79 | 1.41 |
| $Ta_{06} = 7.060 + 0.242 \times LSTd + 0.384 \times LSTn - 0.002 \times elevation$ | 0.80 | 1.38 |
| $Ta_{07} = 8.440 + 0.241 \times LSTd + 0.398 \times LSTn - 0.002 \times elevation$ | 0.86 | 1.20 |
| $Ta_{08} = 7.644 + 0.253 \times LSTd + 0.419 \times LSTn - 0.002 \times elevation$ | 0.87 | 1.23 |
| $Ta_{09} = 5.294 + 0.291 \times LSTd + 0.407 \times LSTn - 0.002 \times elevation$ | 0.84 | 1.24 |
| $Ta_{10} = 3.418 + 0.266 \times LSTd + 0.406 \times LSTn - 0.002 \times elevation$ | 0.83 | 1.26 |
| $Ta_{11} = -0.912 + 0.271 \times LSTd + 0.411 \times LSTn - 0.001 \times elevation$ | 0.83 | 1.57 |
| $Ta_{12} = -2.560 + 0.314 \times LSTd + 0.463 \times LSTn + 0.00037 \times elevation$ | 0.86 | 1.67 |

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
