# Peer review of "Estimation of Climatologies of Average Monthly Air Temperature over Mongolia Using MODIS Land Surface Temperature (LST) Time Series and Machine Learning Techniques"

_remotesensing, doi:10.3390/rs11212588_

Round 1

Reviewer 1 Report

The study tried the performance of random forest (RF) approach on estimating monthly mean near surface air temperatures by MODIS LST products and other environmental factors in Mongolia. It’s interesting that with RF, the G2 group performed better than those (G3-G7) having more independent variables. I recommend major revision due mainly to the weak discussion of the MS and some other points that need to be clarified. 1. The study was made with MODIS LST products of 1 km spatial resolution. I wonder if the monthly mean air temperatures differed significantly from a pixel to its neighbor pixels 1 km apart, when there are rapid air flows. 2. The discussion of the MS was nearly the brief repeat of the results and should be improved, focusing on issues such as the underlying mechanism of the performance of RF+G2 over PLS+G2, RMSE of the monthly mean temperatures in the present study and that of daily air temperatures in literatures, etc. 3. There are a lot of studies on the issue of estimating air temperatures with MODIS LST products published in Remote Sensing, attention should be paid to those studies when revising your discussion. Some other specific comments: 1. Line 127-128: ‘RMSE has been reduced from 1.4-3.1 K for c005 to 0.5-0.8 K for c006’. Please check the RMSE of c006, referring to Duan et al. (2019). 2. Some statements are of subjective judgement and should be revised, e.g., ‘This package is the most notable for RF performance’ (Line226), and ‘Good examples for the aforementioned methods include studies done by…’ (Line73), etc. 3. Line 236-237: ‘The main justification for setting 500 trees is that this value is widely used by analysts using the “randomForest” package’. No need to explain the reason of your 500 trees. It’s an empirical value of a RF application to make sure the final output stabilized. 4. Check the value of yearly RMSE of G3 in Table 4 References: Duan et al., 2019, Validation of Collection 6 MODIS land surface temperature product using in situ measurements, Remote Sensing of Environment.

Author Response

Dear Reviewers,

Thank you very much for your useful comments and suggestions. We have improved our paper according to your comments and made all the corrections. Below, we specify all the changes we made. We also revised the entire manuscript again to improve figures and writing.

The most important changes are:

We refined the abstract section

We refined the introduction section

We refined the study area section

We refined the data and methods section

We refined the result section

We rephrased discussion and conclusion

Figure 1a, b were changed

Figure 3 was changed

Figure 7 was changed

Figure 8 was changed

Figure 9 was changed

Figure A2 was changed

Table 2 was changed

With kind regards,

Ms.O.Munkhdulam

Reviewer 2 Report

Paper “Estimation of climatologies of average monthly air temperature over Mongolia using MODIS land surface temperature (LTS) time series and machine learning techniques” deals with a usage of methods of machine learning in estimating of spatial distribution of average monthly and yearly near-surface air temperature, which can be useful for the estimating spatial distribution of temperature in the large areas with the sparse placed meteorological stations, as authors suggest. It also proves that usage od machine learning algorithms gives more reliable and more precise results than linear modelling techniques.

The title of paper reflects the research theme, the introduction provides sufficient background, research design is appropriate and the methods are adequately described. Although the conclusions supported by the results, there is room for more elaborate presentations of the results. Especially, Figures 7 and 8 should be analysed more detailed and in more aspects. The impact of six land cover types on difference of monthly and seasonal average estimated Ta between RF and PLS regression models are presented but there is no explanation of the reason why this analyses were done, nor what are the impact of vegetation types on studied case.

The comments/suggestions on the paper that should improve quality of the paper are:

Line 6: There are no affiliation of the 4th author.

Lines 73-74: I think that proper way of citing the papers in this lines should be: … done by Zhou et al. [36], Janatian et al. [37]…. In addition, Chak et al. (2014) does not match to the reference no. 38 in the Reference chapter (lines 547-548). There is no full reference of Chak et al. (2014) anywhere in the manuscript.

Line 84: It should be: For instance, Kilibarda et al. [31] estimated…

Line 101: The locational information should be stated, as the terrain parameters are.

Line 105: Why is administrative division relevant information to this research?

Line 107: Figure 1(b) does not show extreme continental conditions with semiarid and arid regions but it shows Köppen-Geiger climate classification.

Line 113. Instead “In contract” it should probably be “In contrast”.

Figure 1(a): Names of aimags should be transcribed from Cyrillic letter to a Latin.

Figure 1(b): In the title of the figure there should be period to which presented Köppen-Geiger climate classification of Mongolia is based on.

Figure 2: In the title of the figure there should be period to which presented climate charts are based on.

            It should be: Climate graphs (charts?) of the three most prevalent climatic zones in Mongolia…

            It is not clear which the source of data used for that figure is. To which "climate data organization" are you referring to?

            The name for the Dwc climate type in this figure is “cool continental and subartic climate”, but in the Figure 2 it is “monsoon-influenced subartic climate”. I suggest usage of the same name for the same climate type.

Line 128: Measuring unit should be °C instead K.

Line 150: What does (Homepage of the R project) refer to? Web-page?

Lines 165-166: How the average air temperature from the 3-hourly meteorological data was calculated? As arithmetic mean of all 3-hourly daily temperature. Since average daily temperature is usually calculated by using temperature at 7, 14 and 21 hours, any different method should be clearly stated.

Line 174: … for the period 2002-2017.

Line 180: You don’t have to use word see. (Table 1)

Lines 189-191/Table 3: It is not clear how the predictor variables were grouped into mentioned groupings. Does that depend on certain values of correlations or is it, at least in part, arbitrarily?

Line 288: Instead Figure 6, it should be Figure 5.

Line 290: The Ta was strongly depended on elevation from April to October, and not only in summer season.

Lines 317-318: Why do you said that in the remainder of the study groupings G1 to G7 were skipped? Does that refer only to a PLS models or on general?

Lines 338-340: It should be: Figure 7(b) showed…. (without In). The sentence: “Figure 7(b) showed that intra-annual analysis or agreement/disagreement between monthly averaged measured and estimated Ta values for 2002-2017.” is likely unfinished.

Line 343: It should probably be: Large discrepancies occurred in transition months such as the start/ the end of seasons. (without - water and tree cover)

Lines 336-343: Whole section seem to be unfinished. There is no proper analyses of Figure 7. There is stated that “the predicted air temperatures show a strong dependency with the elevation (Figure 1), but reveal additional detail and information.” Though there are just few details that are mentioned across that lines. Additionally, in Figure 7(c) example zones were selected in a way to represent six land cover types. Unfortunately, there is no explanation why that was done. How different types of land cover impact the difference of monthly average estimated Ta between RF and PLS regression models? Are there any regularities that can be observed? The analyses of Figure 7 should be more elaborate.

Figure 7: (title) (b) Intra-annual analysis of (blue) coefficient of determination (R2) and (purple) root mean square error (RMSE),

Lines 346-348: (Without “In” at the beginning of the sentence.) Again, only few lines (one sentence) is dedicated to Figure 8 (I have to asks again if this part of text finished.).  Although elaborate analyses of Figure 7 could shorten analyses of Figure 8. At least short comparison to previous figure has to be made and some conclusions given.

Figure 8. On figures 8(b) and 8(c) within y-axis titles should be “seasonal” instead of “monthly”. Again there are no analyse of Figure 8(c). The titles of the figure should be changed (suggestion): (b) Seasonal analyses of (blue) coefficient of determination (R2) and (purple) root mean square error (RMSE), (c) Difference of the estimated seasonality average Ta between RF and PLS regression models by cover type.

Line 351: What does mean “Similar good results”? More reliable, or more accurate results?

Figure 9: “(Bottom) comparison between measured and estimated annual average Ta.” On that figure there are also comparison between measured and estimated monthly average Ta.

Line 361: It should be: …study Noi et al. [39], Kilibarda et al. [31] reported…

Lines 369-375: In the discussion the importance of LSTn and LSTd are mentioned and explained, but the seasonality of residual errors should also be explained, i.e. why the warmer months were generally better modelled compared to extremely cold winter months.

All relevant changes in results or discussions should be mentioned in the Conclusions. Additional attention should be given to the citations, especially to the citing in the text. 

Author Response

(The authors gave the same response as above.)

Reviewer 3 Report

General comment

The purpose of the study is very interesting. From a technical point of view, the use of more and more complex regression techniques, even non-linear ones, and the use of a meaningful number of variables to compute temperature are interesting and the study seems to have a sound motivation. Its execution is well-planned and well-performed, the results robust.

However, the "purpose of the study/the authors’ intentions" are not stated clearly enough in the Abstract and Introduction. The Introduction sounds confusing at times, especially since it does not mention the importance of weather stations for the validation of regression methods, something that is at the core of this same study; only by reading the Methods paragraphs is a reader able to dissipate this ambiguity. Although the Mongolian Ta maps and the final results are a nice addition to the paper, the core of the study seems to be the use of stations for training/validation of the two regression methods and their comparison, so if my understanding is correct it needs to be emphasized more also in the Abstract. A more focused description of purposes and motivation is needed. And a more thorough depiction of the use of weather stations for the validation is needed. 

Check for instance Line 87: “The performance of STRK to predict Ta from MODIS LST products was compared with the performance of the linear regression model”; after this line, the authors talk about RMSEs, but it’s not clear what they have been compared to. We are talking basically about validation studies that compare the performance of different regression methods to build TA from LST, but what are they compared to? Station data, I guess, otherwise what would you compare error against? This is not clear in the text, and the introduction in general gives a skewed impression of what you are trying to achieve.

Along the same lines, it might be appropriate to mention the limitations of satellite based models, for instance spatial and temporal problems, otherwise one could jokingly ask why we are not relying completely on satellite date and waste money and time measuring temperature with AWSs. The main problem mentioned is only that satellites are not measuring directly TA “but only the land surface temperature”, and this doesn’t capture the main problems of satellite retrieval of temperature data (or atmospheric data in general).

Section 3.1. Remote sensing data is, in my opinion, unstructured, with a lot of unnecessary details, that makes it difficult to read. I would suggest a complete re-write. For instance, some questions that popped in my mind when reading were: if data are used from 2002 to 2017, why is a method used only since 2016 discussed at length? Is this 006c data used? Or the 005c data used? Or both, combining one until 2016 and one after 2016?

Another important point that is not addressed is the one referring to R2 values, although some valuable information is implicit in the text. It is known that, given an ever-increasing number of variables, it is possible to get higher and higher values of R2, without there being a real increase in its predicting value. The fact that the authors highlight how only three variables are enough to get meaningful results, is a good qualitative assessment of predicting power, as the high correlation and R2 values and small RMSE errors are more ‘valuable’ since they come from only three variables. It’s easy to find references to this in literature and just a short explanation would suffice to address this point that would give more robustness to the authors’ findings.

It is also interesting that the authors used latitude and longitude as variables. While the use of latitude makes sense, given the different solar radiation absorbed at different months throughout the year at different latitudes, the use of longitude is meaningful probably because it actually is a good proxy for the distance from the sea, which can act as a mitigating factor (studies have shown this in several cold regions in northern latitudes, for instance Norway, Sweden and Finland). I think that the authors would make the use of longitude more robust if they made this correlation explicit. As I am not familiar with the climate of Mongolia and north-east Asia, I might be wrong in this assumption; but as the use of longitude is slightly puzzling me, if my understanding is correct it should be included briefly (maybe with one or two references), if my understanding is wrong then I would really like to know the rationale behind the use of longitude as a predictor in this study. Note that longitude, although not as important as LSTd LSTn and latitude, has nonetheless good predictor values, so mentioning it in the text might be valuable, although not absolutely necessary.

From the language point of view, the paper is clear enough for a non-English native speaker like me in the first half of the study, although several typos are present and some expressions don’t really sound right. I have tried to point out as many as I could later in this review, but this became tougher and tougher farther along in the reading, as the second part of the paper is not linguistically on par with the first half and many sentences start becoming increasingly unclear or plain wrong. This makes the reading unpleasant when the reader gets to the Discussion. A check with a native speaker collaborator or with a professional company might help out with the minor issues, but a thorough check (and some re-writing) is needed for the second part.

Minor comments:

Check “ The LST MOD11A2 (collection 23 6) products provides”

“The PLS regression result showed a coefficient of determination (R2) between measured and estimated monthly Ta climatologies between 0.74 and 0.87, 30 and root-mean-square error (RMSE) from 1.20°C to 2.19°C.” --> re-write

“Ta at height 2 m above the land surface” --> above the surface/ground should suffice

Line 33 “ the two best modeled months were July and August, and the two worst months January and February.”

Line 66 “related for example to physical properties and atmospheric conditions” --> unclear, expand upon

Some inconsistency in the citations, compare line 49 page 2 with lines 73-74 page 2

The list at the end of page 2/start of page 3 (Lines 80-81) is inconsistent: why “geographically  weighted regression”/”generalized boosted model” and then “stepwise”/”ordinary least squares”?

83 “Multivariate and non-parametric algorithms and RF and STRK have been highlighted as the foremost algorithms to ensure high accuracy.” --> re-write

Line 98: “Our research objective was to develop a robust empirical models to estimate climatologies of average monthly and yearly Ta 99 across Mongolia at 1 km spatial resolution using time-series of MODIS Terra LST products, terrain 100 parameters (elevation, slope, aspect) and locational information.” --> Not sure about the use of the term “locational”. Moreover correct “to develop a robust models” --> “a robust model”.

Page 3, Line 103: “The study area covers the entirety of Mongolia with a total territory of approximately 1.56 × 106 103 km2.” --> I’d change territory with area

Page 3, Line 104: “It is sub-divided in 21 administrative units, and its land surface elevation is between 524 m and 4320 m above sea level in Figure 1(a). The climate conditions are extreme continental with semiarid and arid regions Figure 1(b). The “blue sky” country counts on average 260 sunny days per year and is characterized by long-cold winter, and a short dry-hot summer, with generally low precipitation” --> several typos, check

Page 3, Line 112: “In Köppen-Geiger’s climate classification, Mongolia is involving five different classes [91] 112 (Figure 1(b)).” --> this sentence doesn’t really sound right. Maybe you could re-write this as “According to the K Köppen-Geiger climate classification, Mongolia can be divided into five climatic regions.” Check also typo in line  113, “in contract” should be “in contrast”.

Page 5, Line 127: “For instance, RMSE has been reduced from 1.4-3.1 K for c005 to 0.5-0.8 K for c006.” --> How is RMSE computed? RMSE compared to what?

Page 5, Line 136:  “For this study, used Level-3 MODIS Terra 8-day LST product (MOD11A2)” --> subject is missing

Page 5, Line 150:  “(Homepage of the R project).” --> ?

Page 9, Line 215: “The RF algorithm provides out-of-bag error (OOB)” --> out of bag error is referred to several times, but only explained in Page 10, Line 239-244. I think it would be beneficial to the readers if the authors could find a way to organize the material referring to OOB differently.

Page 10, Line 235: “The main justification for setting 500 trees is that  this value is widely used by analysts using the “randomForest” package” --> This is not an acceptable justification, at least try to smooth up this assertion, for instance  re-write along lines like “It is difficult to determine a correct number of decision trees, however many studies using the Random Forest method agree on about 500 (source 1; source 2; source 3)”

Page 10, Line 246: “For comparison – and to assess the differences between linear and non-linear models”, then at line 249  “allows us to quantify the benefits of using a non-linear machine learning technique (RF) compared to a parametric linear approach (PLS)”, one of the two sentences is redundant. Better to combine them or to take away one of them.

Page 11, Line 270: “The VIP values determine the contribution of the predictor variables to the PLSR  latent variables and are generally set greater than 0.80” --> not sure what “are generally SET greater than 0.80” means. I suppose that the authors want to say that values greater than 0.80 identify the most important variables (like they say later on in the paragraph)

Page 13, Line 307 “vaiables” –> “variables”

Page 14, Line 338: “In Figure 7(b) showed that intra-annual analysis or agreement/disagreement between monthly averaged measured and estimated Ta values for 2002-339 2017.” --> not clear

Page 14, Line 340: “ The large difference was also observed during the colder months (December to March). It also demonstrated that inner-annual distribution of difference of monthly average estimated Ta between RF and PLS regression models (Figure 7(c)).” --> Another unclear sentence. What is the subject of “It”? And what did it demonstrate? It demonstrated that the distribution... No verb is given here.

Page 14, Line 343: “Large discrepancies occurred water and tree cover in transition months such as the start/ the end of seasons.” --> This sentence makes no sense

Page 17, Line 346: “In Figure 8 showed that spatial distribution map of the estimated seasonality average Ta (Figure 8(a), models error (Figure 8(b), and difference of both models (Figure 8(c) using LSTd, LSTn, and elevation as predictor variables.” --> This sentence is not correct. I suppose what the authors are saying is: “In Figure 8, the spatial distribution map of the estimated seasonal average Ta is shown (Figure 8(a)), along with model errors (Figure 8(b)) and the difference between the models (Figure 8(c)). Predictor variables were LSTd, LSTn and elevation”

Page 17, Line 356 “RMSE of about 0.84-1.93°C” --> I think that a mention of the relative error might also be relevant to show the accuracy of the study. An absolute measure is not enough, especially for readers like me that are not aware of general temperature patterns and diurnal, monthly, yearly temperature variations in Mongolia. A line or two to mention this would suffice.

Page 17, Line 358: “Without any hyperparameter tuning, the non-linear RF models outperformed linear PLS models confirming other studies” --> “confirming other studies” should be re-written better, into something along the lines of “outperformed linear PLS models, in accord with other comparative studies ....” and here you might mention whether the studies you refer to deal with PLS vs RF always and in general, or PLS vs RF in temperature reconstructions.

Page 17, Line 360: “ In a study in Noi et al (2017) [39], Kilibarda et al [31] reported also a high importance of day and night-time LST observations as well as elevation.” --> Another confusing line. Kilibarda et al reported in a study in Noi et al?

Page 18, Line 372: Another broken sentence: “Day-time land surface temperature were nevertheless found important, as LSTd significantly impacts the latent heat flux of Ta [40], respectively, shows strong interdependencies with this water vapor flux as a high evapotranspiration has a cooling effect on vegetation.”

Page 18, Line 390: “The method is relatively easy to implemented” --> implement

Conclusions:

I think the study is interesting and I recommend wholeheartedly its publication after these minor issues are solved. Although I think there will be some parts that need to be re-written and a few comments and citations that need to be added, these are mainly presentation issues, minor details, or English language problems, so they are not marring the achievements of this study. The main goal and results of the paper are interesting and the paper has been an enjoyable enough read, after everything has been taken into account.

I want to thank the Authors for their effort, and I want to thank the Editor for having provided me with the opportunity to read, comment, and hopefully improve upon this study.

Kind regards.

Author Response

(The authors gave the same response as above.)

Round 2

Reviewer 1 Report

The revised MS responded carefully to all the referee’s comments. Though some points were not discussed thoroughly, I don’t think they make major obstacle to the acceptance of the MS. I recommend minor revision concerning the following points:

Line 229~230: “Variable importance is measured by computing the increase in mean square error (MSE) when the OOB data for each variable are replaced [60,70]”. Replace by what?

Line 232~234: “Our analysis computed two kinds of variable importance measures using the “randomForest” package in R [63]: (1) percent increase in the mean square error (%IncMSE) and (2) increase in node purity (IncNodePurity).” I didn’t find any statement of IncNodepurity in Result or Discussion sections of the MS.

Line 379~381: “The contours of Köppen-Geiger and Köppen’s climate maps are broadly reflected but obviously, lack the information on precipitation. Information about precipitation would be needed in order to refine the Köppen-Geiger climate maps”. The present study didn’t tap precipitation, so it’s irrelevant to say about precipitation.

Author Response

Dear Reviewers,

Thank you very much for your useful comments and suggestions. We have improved our paper according to your comments and made all the corrections. We also revised the entire manuscript again to improve writing.

With kind regards,

Ms.O.Munkhdulam

Reviewer 2 Report

Thank you for a very detailed response to my review. I find that the paper is significantly improved compared to an original version. The analyses of the results is more elaborate and concentrates on the results that are relevant to a topic of the research. The impact of the land cover type on the differences between RF and PLS regression models could be a topic of a further research. I suggest to accept the paper in a present form, but after any comments from other reviews are considered. As English is not my native language, I suggest language review, if that is necessary.

Author Response

(The authors gave the same response as above.)
